# New Generation of the Compact System for Performing Measurements of Sold Liquids by Gas Station Dispensers

Jaromír Markovič [1,*], Jozef Živčák [2], Milan Sága [1] and Pavol Tarbajovský [2]

1 Faculty of Mechanical Engineering, University of Žilina, Univerzitná 8215/1, 010 26 Zilina, Slovakia; milan.saga@fstroj.uniza.sk
2 Faculty of Mechanical Engineering, TU Košice, Letná 9, 040 01 Košice, Slovakia; jozef.zivcak@tuke.sk (J.Ž.); pavol.tarbajovsky@tuke.sk (P.T.)
* Correspondence: jaromir.markovic@fstroj.uniza.sk

**Abstract:** This paper presents an advanced compact system that represents an innovative solution determined for the precise measurement of the fuel liquids and chemical materials used in the transport area. This system was created as a product of the applied research, development, and following realisation in the Slovak Legal Metrology. This organisation is authorised for the certification of the measuring systems. The given system enables to perform metrological control for a wide range of the measuring systems in order to achieve reliable results in the measuring process and the required measuring precision, as well as to minimise idle times of the measuring stands during the metrological control. In addition, the presented system is user-friendly and it requires only a short time for training of the metrological personnel.

**Keywords:** compact system; sold liquids; metrology; gas station dispensers

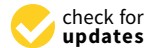

## 1. Introduction

Dispensers used for volume measurements of sold liquids at pump stations can be sorted as measuring systems for the continuous and dynamic measurement of amount of liquid, except water. Measurement systems' metrological and technical requirements for continuous and dynamic liquid amount measurements except water, which undergo legal metrological control, are stated in international recommendations OIML R 117-1 [1,2]. International recommendations contained in OIML R 117-1 are used as a normative document within conformity assessment of dispensers offered and in the EU market according to Regulation 2014/32/EU for measuring instruments [3]. OIML R 117-1 includes metrological and technical requirements for diesel, gasoline, LPG, AdBlue, washer fluid dispensers, etc.

Automation is becoming commonly used in industry practice. Automation benefits extend to the field of metrology in the form of rejuvenation for often repetitive processes and are supported by the credibility of measurement results. Automation trends penetrate all industrial branches including metrology. The idea of automation application in the field of metrology control of gauges used at pump stations comes from practice requirements and activities performed by the Research and Development department of Slovak Legal Metrology. Slovak Legal Metrology as the main partner of this project has cooperated with prominent Slovak technical universities.

The main purpose of this article is to introduce a new generation of the compact system for performing metrological controls (mainly confirmation) of dispensers used at pump stations (AMSV). Dispensers are sorted as dynamic measurement systems for liquids except for water. There are several ancestors of this system which differ mainly by the considerably higher level of automation and wider range of provided services [4].

The goal of the project was to create a measurement system that would be able to check all dispensers available at pump stations with the usage of artificial intelligence

elements, such as the automation and acceleration of measurement processes with compact construction with the ability of integration into conventional commercial vehicles up to 3.5 tones.

During the design process, the usage of electric systems in a potentially explosive environment was taken into account concerning the safe use of the AMSV system.

Figure 1 shows the AMSV system integrated into a commercial vehicle, thereby creating a mobile system for measurement. Individual subsystems are described in the following sections.

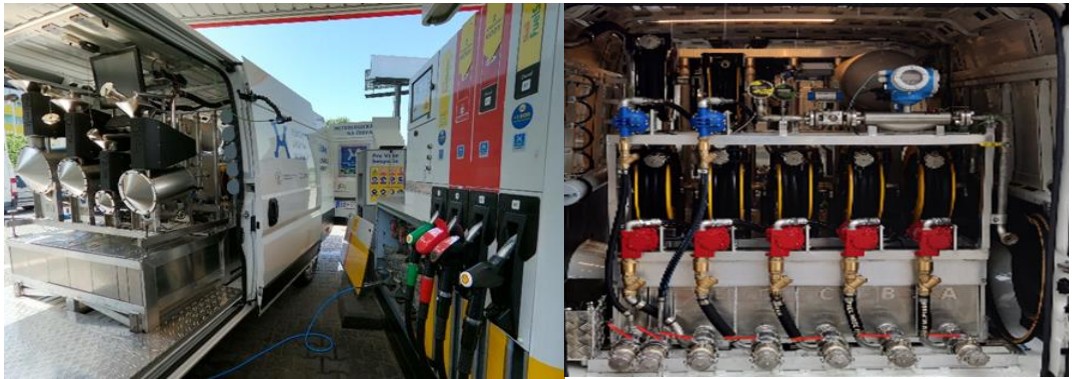

**Figure 1.** The AMSV system as a part of the conventional commercial vehicle up to 3.5 tones.

## 2. Mechanical Part of the AMSV System

It was necessary to consider the whole system weight, taking into account its strength and rigidity during the design process. The development of the compact system, which weighs with an appropriate vehicle and whose driver would not exceed the maximum capacity load (3.5 t) of the commercial vehicle, was the goal of this project. In addition, easy insertion of the whole system into the commercial vehicle cargo space was required without modifying vehicles in a wide range. Such a complex system requires a lot of sensors, tanks, standard containers, pumps, hose systems, etc., for the verification of all types of measuring sets. Due to mentioned reasons, it was not possible to use conventional materials such as steel for all construction elements. The best weight-saving parts, as it turned out, were collection tanks and a system of supporting frames. There are three types of frames: mainframe (Figure 2), frame for LPG technology, and frame for standard measuring containers. Collection tanks and the supportive frame system were made out of aluminum alloy, which has one-third density and one-half the failure strength counter to conventionally used steels. The weight of the whole construction was lowered by 350 kg thanks to this approach. Strength analysis of supporting frames and collection tanks was performed in the form of engineering simulations due to safety reasons. The aim of these simulations was:

- The static load of the system under the action of gravitational force, important during the opening process of full collection tanks.
- The biaxial static load of the system under the action of gravitational force and braking force (proportional to gravitational acceleration 0.8 g).
- The biaxial static load of the system under the action gravitational force and centrifugal force (proportional to gravitational acceleration 0.4 g), considering system behavior at curves.

Simulation results of all measurements proved high coefficients of safety (values higher than 4) [5,6].

A 3D model of the system as a compact device is shown in Figure 3.

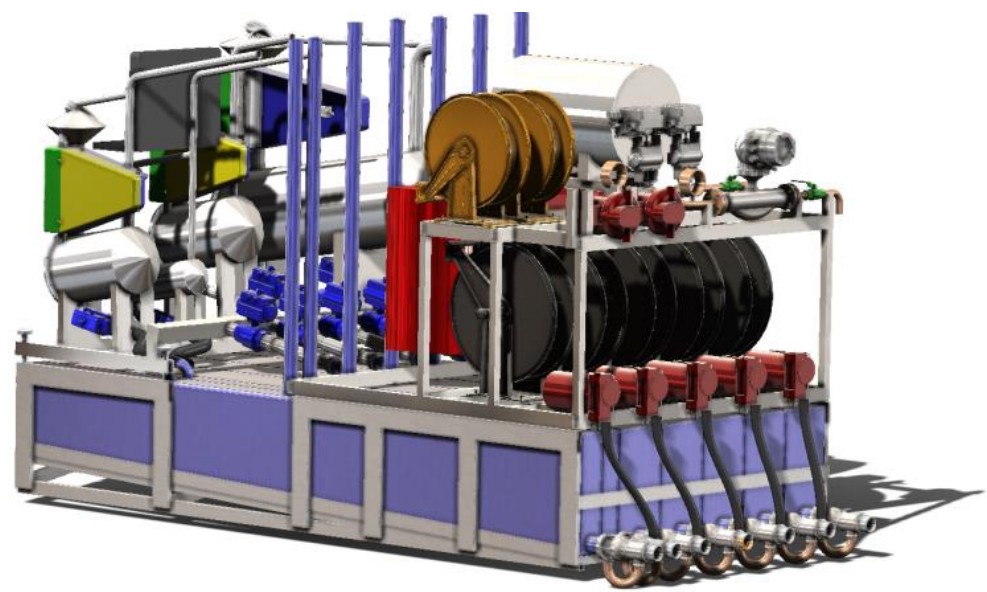

**Figure 2.** The AMSV compact system (3D model).

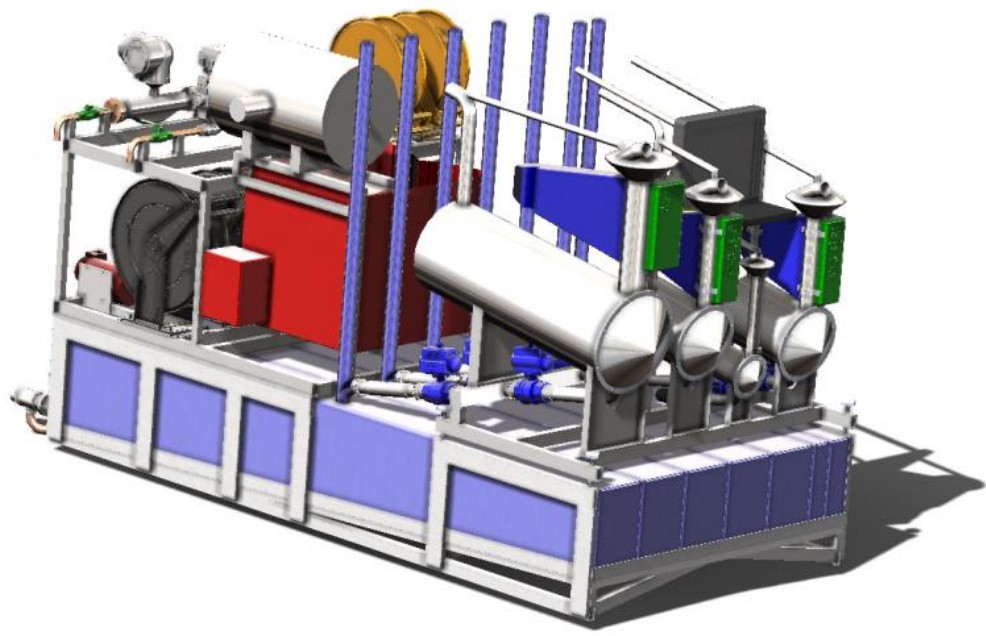

**Figure 3.** The AMSV compact system (3D model).

### 3. Electrical Solution and Software Solution

The electrical system is designed to be independent. The power supply system uses four batteries with a total capacity of 200 Ah at their 24 V power supply. The power supply system includes converters, batteries, and diode switches [7–9].

When powered by an external power network, a 230 V diode switch switches the power supply from the batteries to the invertors and the batteries start to charge. The whole electrical system was designed to cover the 15 h activity of the AMSV system at an overdraft of 6000 L of liquid (cabin temperature of 20 °C) or 15 h of activity at overdraft of 4500 L of liquid (cabin temperature of 0 °C). A simplified system electrical scheme is shown in Figure 4.

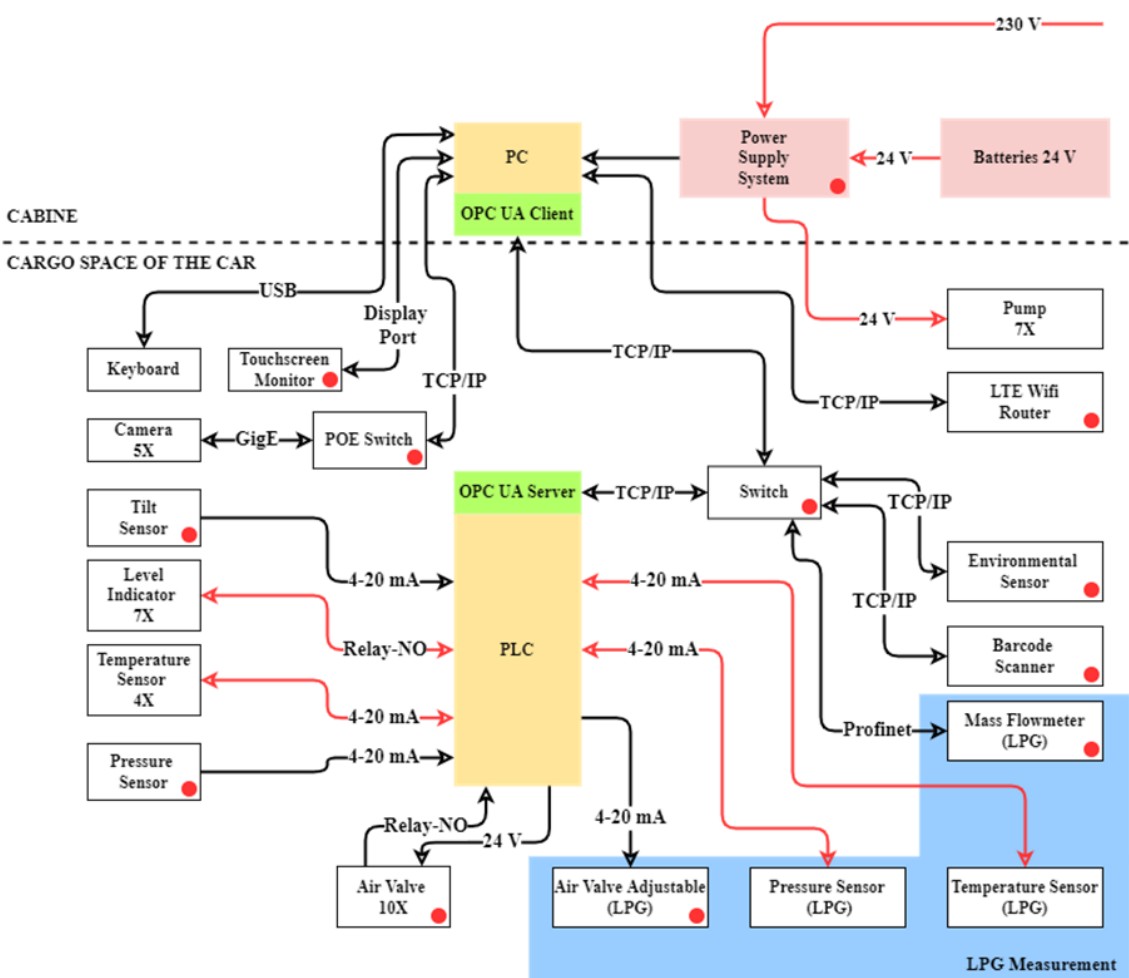

**Figure 4.** Simplified scheme of electrical system.

The AMSV system is controlled by the technical operator with the help of software, which is run on the computer and together with PLC [10–12] forms the management part at the highest level. The OPC UA communication protocol [13,14] serves as a communication channel, where PLC is a server and PC is a client. OPC UA is a safe and independent platform for data exchange between PC and PLC. Control of the pneumatic valves and data collection from all sensors is defined by the mentioned data exchange which brings a high level of automation. PLC implements regulation algorithms and also performs checks of control parameters. The AMSV system uses intelligent image processing (in recognizing indicated dispenser data and in determining the volume of liquid in standard measuring containers). Image processing is performed using cameras that are connected to the PC via a POE switch. A sensor, such as a barcode reader (used for measuring instrument identification from measuring instrument database), environment sensors (temperature, pressure, and humidity), or a mass flow meter, communicates with PLC via TCP/IP protocol [15] through the switch [16]. All used valves have pneumatic propulsion with electric control and use a compressed air tank, in which air pressure is constantly measured.

An electro-pneumatic positioner [17] (used for setting flow during LPG measuring) is controlled by a (4–20) mA current, while START/STOP valves are controlled by voltage with the value of 24 V. Every START/STOP valve contains a pair of valve end position sensors with a relay output (connected as NO). These sensors are used for emptying standard measuring containers into collection tanks or when verifying LPD dispensers. Approximately 90% filling of collection tanks is indicated by the usage of level indicators. With temperature compensated dispensers, the temperature of liquids is measured in standard measuring containers. The biaxial inclinometer is used for the plane setting of

standard measuring containers. A temperature and pressure meter with current output (4–20) mA is used for the verification of dispensers in the AMSV system.

The LTE Wi-Fi router creates a connection for the AMSV system (Figure 5) with the verifier's meter database via the internet with the usage of a GSM connection [18]. This connection can be used for uploading measurement data to the authentication provider´s cloud.

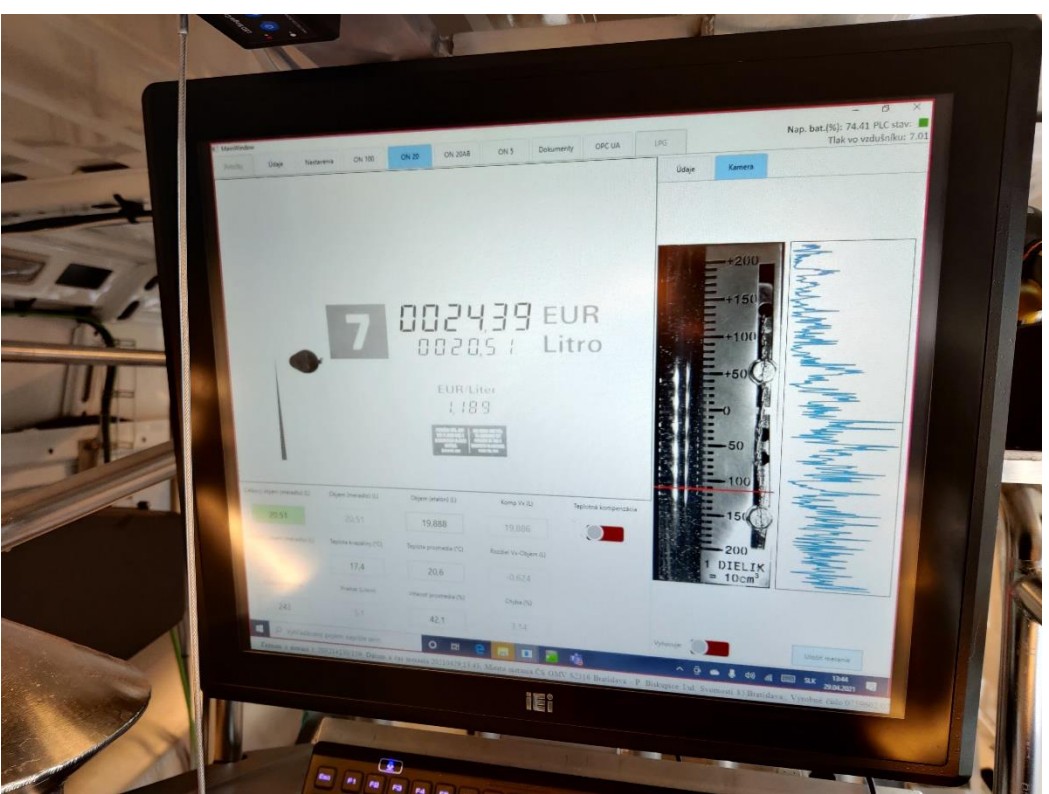

**Figure 5.** The AMSV system software with touch control.

### 4. Measurements

The AMSV system is a complex measuring device that is able to verify all available dispensers placed in a pump station thanks to its equipment. Measuring methods can be divided into two basic groups.

- Liquid volume measuring using standard measuring containers at atmospheric pressure (diesel, gas, AdBlue, and washer fluid).
- Liquid volume measuring using standard measuring containers in a closed circuit.

A high level of automation is used in every measurement.

Liquid volume in standard measuring containers as well as determination of indicated data of dispenser is realized through intelligent image processing.

Measurements that are not based on image processing are controlled by PLC (Figure 4). The following measurements directly affect liquid volume measurement:

- Biaxial tilt measurement of standard measuring containers.
- Temperature measuring of measured liquid (temperature compensated liquid volume).
- Measurement of environment temperature, pressure, and humidity.
- Measurement of the flowed liquid volume using a closed-loop mass flow meter (LPG).
- Closed circuit temperature measurement (LPG).
- Closed circuit pressure measurement (LPG).

Measurements which do not directly affect liquid volume measurement but are important in the process of automation:

- Indication of filled state of the collection tanks.
- Measuring the pressure in the compressed air tank: pneumatic valve factuality testing.
- Battery voltage measurement as a part of the power supply system.
- Monitoring the state of the end position of pneumatic valves.
- Meter barcode reading, for the unambiguous identification of meters in the verification provider´s database.

### 4.1. Liquid Volume Measurement Using Standard Measuring Tanks at Atmospheric Pressure

Measurement is performed by the value method with a stand-still start. The system offers four standard measuring containers with the goal to secure verification of the dispersion in the whole measuring range.

AMSV systems´ standard measuring containers are shown in Figure 6. The design of these standard measuring containers has a big advantage, which is minimal foam formation on the surface of the liquid during measurement. This shortens the waiting time for the right conditions which are needed to measure the received liquid value.

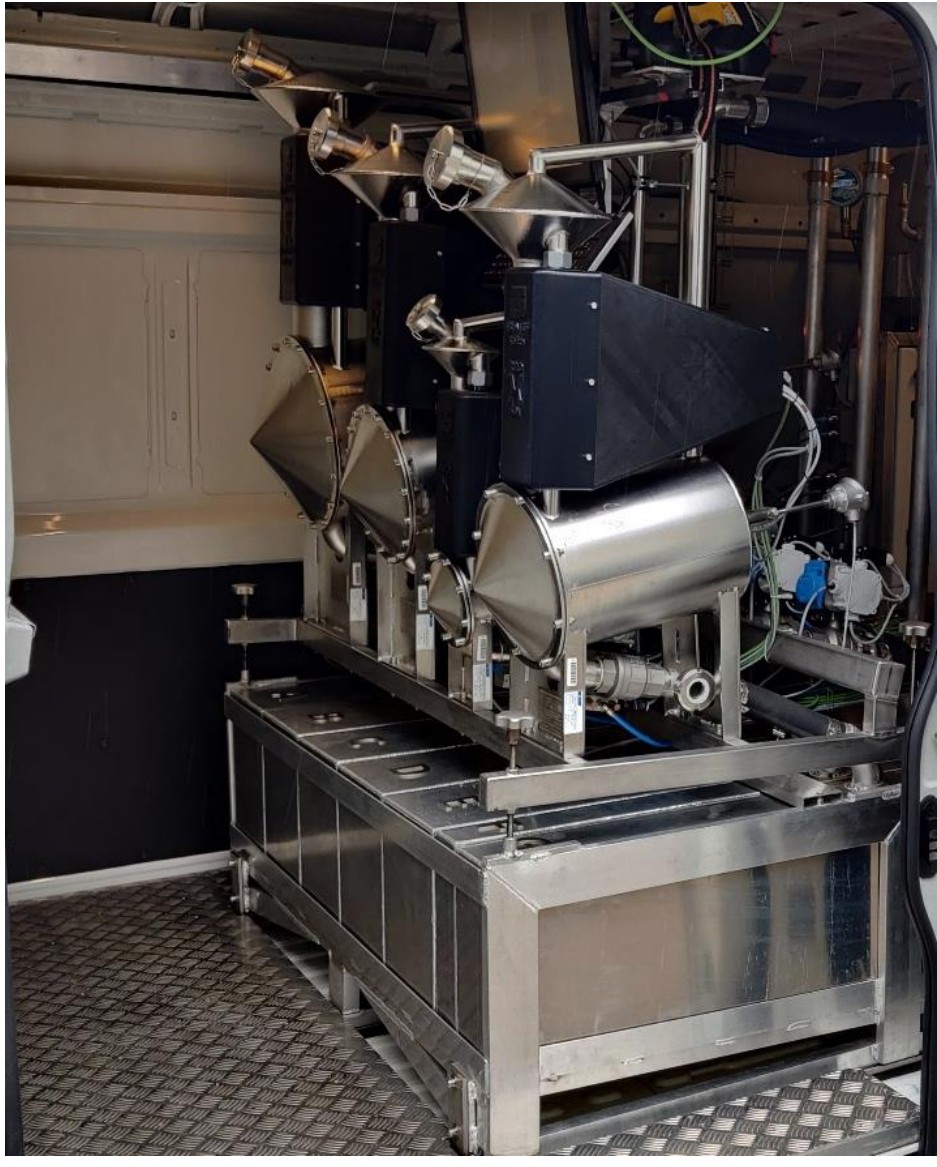

**Figure 6.** Standard measuring vessels used in the new generation of the AMSV system.

Intelligent image processing is used to measure the liquid volume in standard measuring containers (Figure 7).

**Figure 7.** Determination of liquid volume in standard measuring vessels using intelligent image processing.

The camera captures the image of the status indicator, which is then input to the correlation-derivation algorithm. The algorithm evaluates the position of liquid level according to the image and performs a volume recalculation thanks to defined sensitivity. The issue of edge detection and machine vision is generally discussed in [19,20]. The volume is measured continuously in the range of the scale by this method. The value recognized by the algorithm is controlled by the metrological staff.

During the design process of liquid volume measurement, several measurement principles have been considered. The advantage of the chosen principle against other principles is significantly higher measuring dynamics. The radar level meter, for example, needs a certain time to stabilize the indicated value and its electronics are more sensitive to the temperature influence. Visual control of the measurement results is another big advantage of the optical principle (camera) for measuring liquid volume.

Intelligent image processing is also used to read indicated data from the dispenser display. Results digitalization by optical sign recognition, as part of automation, is an appropriate way due to legislation that determines the indicated value shown at the dispenser display as authoritative for payment. Intelligent image processing is the future trend in the field of legal metrology. Image recognition takes place in difficult conditions, which are formed from three factors:

- Dispensers at pump stations are manufactured by different companies with diverse types of indication of the amount of liquid dispensed.

- Surrounding light conditions: when placing a dispenser in the exterior, a wide range of light conditions is formed.
- Dispensers´ displays use plastic or glass protective cover which is the source of light reflection.

Reliable image processing can be in most cases impossible due to reflections created on the protective covers of displays. A POLARSENS chip-equipped camera can filter images in four polarization channels: 0°, 45°, and 135°, resulting in reflection reduction. In the case when some indicated units indicate data in certain polarization, it is possible to choose a different polarization, which does not suppress indicated data of the indication unit. When the reflection causes inaccuracies, the measurement can be repeated.

An image without reflection is then used as input for the OCR algorithm. The OCR algorithm uses elements of artificial intelligence, in particular convolutional neural networks with deep learning [8]. The output is not only a recognized value, but also the probability with which this value was recognized. In this automated process, the technical worker just controls the recognized value.

### 4.2. Liquid Volume Measurement Using Standard Closedloop Mass Flow Meter (LPG Measurement)

Measurement is realized by a mass flow meter [21–24] (Figure 8) using the volume method with a standstill start (Figure 9).

The process of measurement is fully automated once the AMSV system is connected to the LPG dispenser. The technical staff sets required flows and volumes (Figure 10 shows an example of software part on which LPG verification is based) necessary to perform measurements.

The verification process is autonomously controlled by PC and PLC in the following steps:

The regulation process of required fluid flow was achieved by an electro–pneumatic positioner and controlled by a regulation algorithm implemented in PLC [25,26].

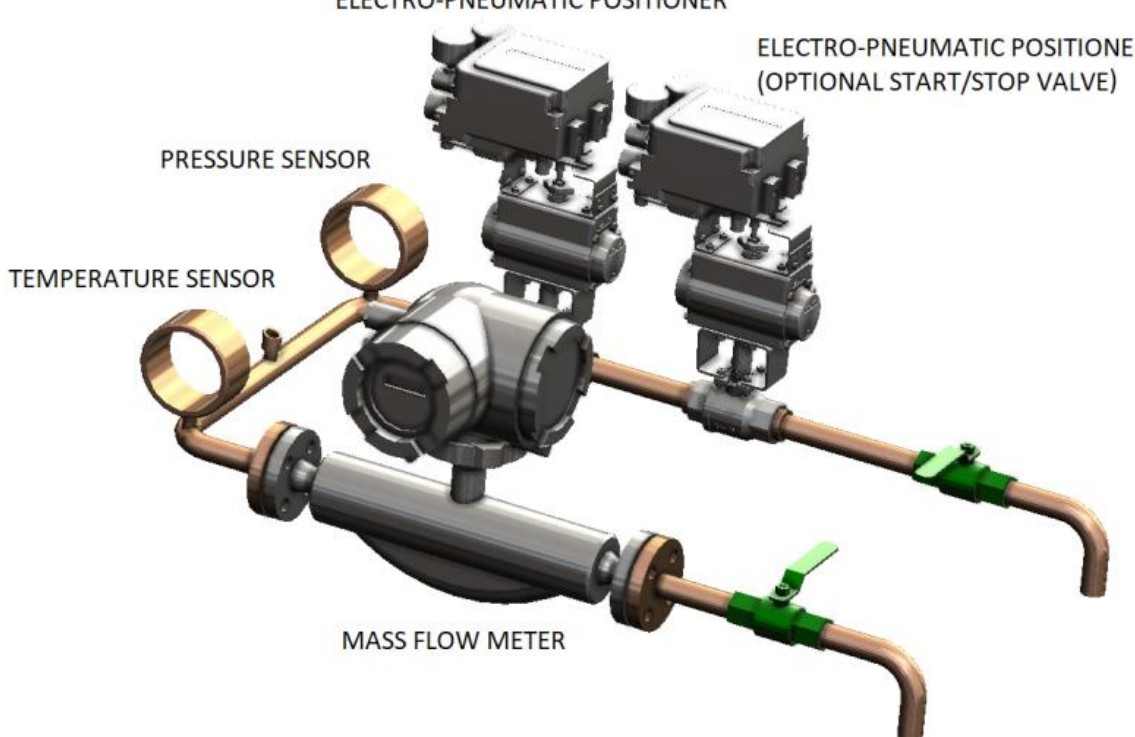

**Figure 8.** Standard measuring set for measuring the overflow volume of LPG.

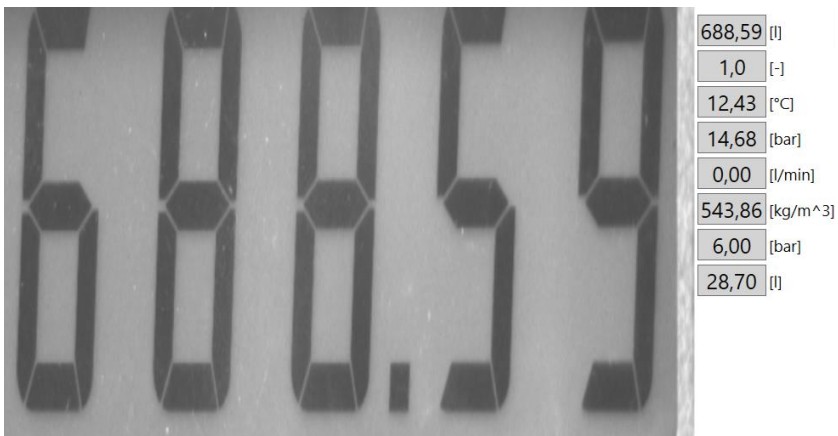

**Figure 9.** Software user interface when verifying the LPG dispenser.

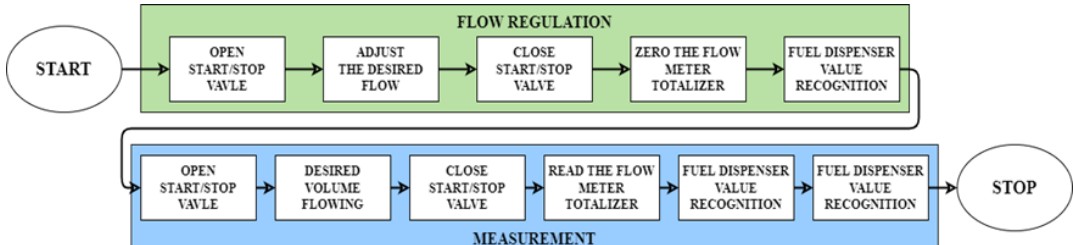

**Figure 10.** Fully automated LPG verification process.

## 5. Credibility of the Measurement Results

The credibility of the measurement results is an important factor.

The AMSV system uses three levels of results credibility control:

- Intelligent data processing: technical staff is not the only one who reads the data from the meters.
- In the case of changes in data recognition by technical staff, this change is recorded.
- Creating a so-called witness photo (Figure 11), where all key information of measurement is captured: dispenser indicated data, a level indicator of the standard measuring container, compensated values, time and place of measurement.

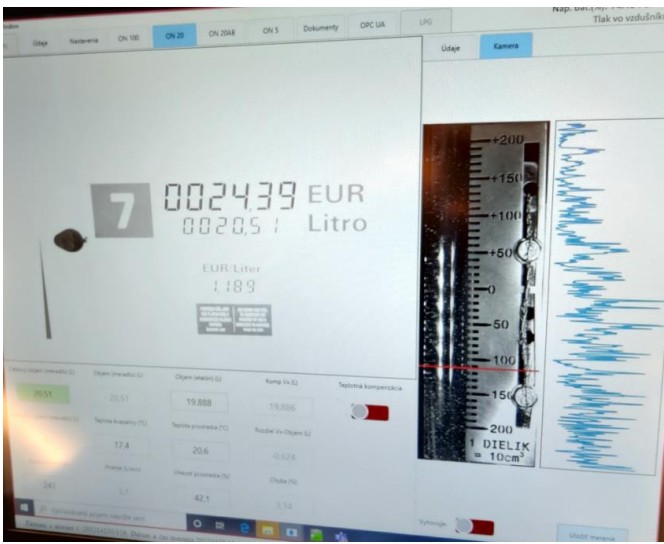

**Figure 11.** Witness photography as a check of the reliability of measurement results.

A witness photo of the measurement is shown in Figure 11.

## 6. Technical Specification of AMSV System

The AMSV system allows for the verification of dispensers available as standard at pump stations with the maximum flow to 200 L/min [27]. These are dispensers for:

- Standard hydrocarbon fuels (without temperature compensation).
- Standard hydrocarbon fuels (with temperature compensation).
- Liquefied natural gas (LPG).
- Washer liquid.
- AdBlue/uric acid.

The outcome of metrological control is composed of the following documents:

- Verification certificate.
- Calibration certificate.
- Confirmation of the liquid volume used during metrological control.
- Confirmation of performance.

The AMSV system has four standard containers, the characteristics of which are listed in Table 1.

**Table 1.** Metrological characteristics of AMSV system.

| Standard Container | ON5 | ON20AB | ON20 | ON100 |
|---|---|---|---|---|
| Specification | Washer liquid | AdBlue | Hydrocarbon fuels | Hydrocarbon fuels |
| Nominal volume (L) | 5 | 20 | 20 | 100 |
| Measuring range (L) | 4.95–5.05 | 19.8–20.2 | 19.8–20.2 | 99–101 |
| Volume of collecting tanks (L) and their number | 90 (1×) | 110 (1×) | 240 (5×) | |

The expanded measurement uncertainty complies with the requirements of OIML R 117-2 [28].

Once the volume measurement in the standard measuring container is completed, the liquid can be moved to a specific collection tank and the container can be reused. In the case of measuring hydrocarbon fuels, both standard measuring containers (ON20 and ON100) are connected to five collection tanks, which allows for the sorting of individual liquids (petrol diesel), as shown in Figure 12.

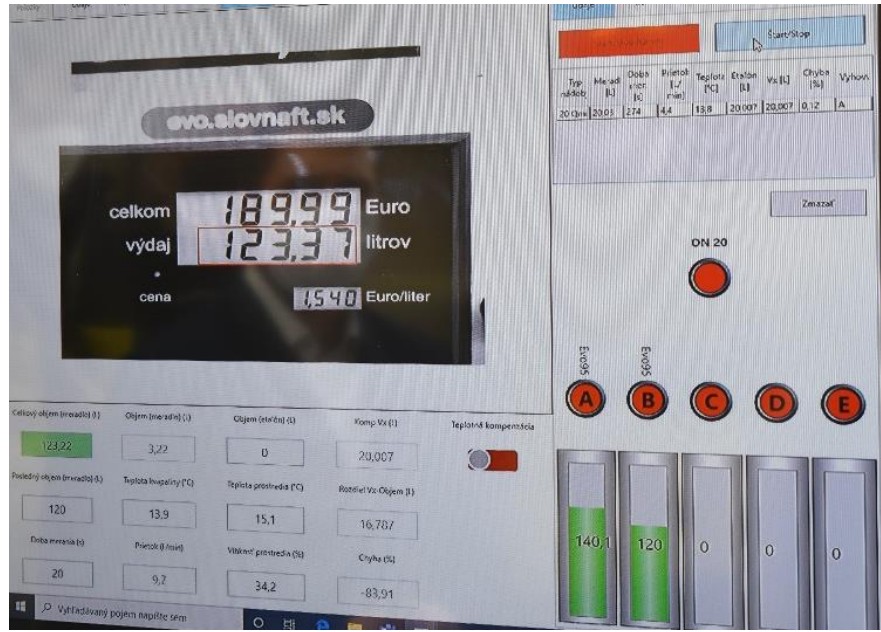

**Figure 12.** Method of sorting liquids into separate collection tanks (control software screenshot).

The collection tanks of hydrocarbon products can be drained using pumps with a flow rate of 50 L/min and the collection tanks for washer fluid and collection tanks for AdBlue can be drained with a flow rate of 30 L/min [22].

Other technical data are given in Table 2.

**Table 2.** Technical data of the mobile AMSV system.

| AMSV | | | Characteristics |
|---|---|---|---|
| Batteries 24 V | External ~230 V | Recharging YES | **Power supply** |
| Operating * | | Charging | **Temperatures** |
| −20 °C ~ +60 °C (outside temperature) | | −10 °C ~ +40 °C (cabin temperature) | |
| | Volume | | |
| 240 L | 110 L | 90 L | |
| | Number | | |
| 5 | 1 | 1 | **Collection tanks** |
| | Pump | | |
| 50 L/min | 30 L/min | 30 L/min | |
| Approximate time of depletion of the entire tank | | | |
| 5 min | 4 min | 3 min | |
| GSM, Wi-Fi | | | **Connectivity** |

* Temperature defines trouble-free usage of equipment, but the temperature at which metrological control can be performed can be modified by a special regulation according to the valid legislation.

## 7. Conclusions

Dispensers are one of the most used types of measuring instruments in business relationships worldwide. This fact was a motivation for the development of a new system generation for the metrological control of dispensers used at pump stations (AMSV system). The goal was to develop and manufacture a measuring system that would allow for realizing the verification of dispensers used at pump stations to increase credibility and efficiency, i.e., minimizing the downtime of dispensers as a source of financial loss of the seller. As mentioned in the article, the AMSV system uses automation in the form of intelligent image processing, complex data collection, monitoring of accompanying quantities, and their evaluation for completing the established goal.

The system has undergone complex tests and is currently used in the Slovak Republic.

**Author Contributions:** Conceptualization, J.M. and J.Ž.; methodology, J.M. and J.Ž.; investigation, J.M. and M.S.; data analysis, J.M. and M.S.; writing—original draft preparation, J.M. and P.T.; writing—review and editing, P.T. All authors have read and agreed to the published version of the manuscript.

**Funding:** The article was written in the framework of Grant Projects: VEGA 1/0318/21 "Research and development of innovations for more efficient utilization of renewable energy sources and for reduction of the carbon footprint of vehicles", KEGA 006TUKE-4/2020 "Implementation of Knowledge from Research Focused on Reduction of Motor Vehicle Emissions into the Educational Process".

**Institutional Review Board Statement:** Not applicable.

**Informed Consent Statement:** Not applicable.

**Data Availability Statement:** Not applicable.

**Conflicts of Interest:** The authors declare no conflict of interest.

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
