# Peer review of "New Generation of the Compact System for Performing Measurements of Sold Liquids by Gas Station Dispensers"

_jmse, doi:10.3390/jmse10040524_

Round 1
Reviewer 1 Report
The reviewed article is very interesting and it is suitable for the scientific publication. It represents an advanced system designed for the area of transport with the aim to perform an accurate measurement of the liquids and chemical materials. I really appreciate the practical importance of the publication, which is not remaining only in the theoretical level, but it has found its practical application in a certified organization. I have the following comments concerning this publication:
1. It is necessary to supply more references and also to add the citations into the text.
2. There are missing citations concerning the figures and tables in the text(figure 8, 11,12-14 ...).
3. The figures Fig. 4 and Fig. 9 are unreadable; it is necessary improve their quality.
4. The opponent sees a possible risk of reflecting the daylight on the camera, which can cause an inaccuracy of the measuring process. How do you intend to eliminate this problem?
Author Response
Dear Reviewer, we really appreciate your time and help with our article. We tried to cover all your comments into the article. All changes are highlighted in the text.
- References have been supplemented
- Missing citations have been supplemented
- Picture quality have been improved
- When this problem occurs, it is possible to eliminate it by measurement repetition.
Reviewer 2 Report
The paper shows a technological development where "The main purpose of the article is to introduce a new generation of the compact system for performing metrological controls (mainly confirmation) of dispensers used at pump stations (AMSV)", and "...The goal of the project was to create a measurement system, which would be able to check up all dispensers available at pump stations with the usage of artificial intelligence elements as automation and acceleration of measurement processes...".
However, in the abstract, we can read ..."The scientific publication introduces an advanced magneto-optical system, which represents an innovative solution determined for precise measuring of the fuel liquids and chemical materials used in the transport area...".
Therefore: there is a distance between the title and the content of the paper; there is not a detailed analysis of the applied artificial vision methods approximation, characteristics, and this goodness.
It is necessary to change the title and/or improve the applied artificial vision methods approximation, characteristics, and this goodness.
It is recommended to contrast your results with other ones.
Must include scientific literature relative to the employed analytic methods.
Author Response
Dear Reviewer,
Let me express my cordial thanks to you for your amazing support, which you have shown me in reviewing of my article. Thank you very much for your time and help, we appreciate it. The title of the article with abstract has been changed and also scientific literature has been supplemented. Once again, big thanks for your help and support.
SIncerely
author
Round 2
Reviewer 2 Report
Thanks very much by improve the paper.
In order to enrich the final part of the paper, I suggest reflex about ¿What future research directions do you think can be addressed? ¿What open problems can you report?
Author Response
Dear reviewer, once again thank you so much for your opinion and review, I really appreciate it.
Sincerely
atuhor